# Transcriptional markers classifying *Escherichia coli* and *Staphylococcus aureus* induced sepsis in adults: A data-driven approach

**Mahnaz Irani Shemirani**📖*

Department of Laboratory Medicine, Institute of Biomedicine, Sahlgrenska Academy, University of Gothenburg, Gothenburg, Sweden

* mahnaz.irani.shemirani@gu.se

**Data Availability Statement:** All dataset analyzed during the current study are available from the GEO repository (accession numbers GSE33341, GSE13015, GSE65088).

## Abstract

Sepsis is a life-threatening condition mainly caused by gram-negative and gram-positive bacteria. Understanding the type of causative agent in the early stages is essential for precise antibiotic therapy. This study sought to identify a host gene set capable of distinguishing between sepsis induced by gram-negative bacteria; *Escherichia coli* and gram-positive bacteria; *Staphylococcus aureus* in community-onset adult patients. In the present study, microarray expression information was used to apply the Least Absolute Shrinkage and Selection Operator (Lasso) technique to select the predictive gene set for classifying sepsis induced by *E. coli* or *S. aureus* pathogens. We identified 25 predictive genes, including *LILRA5* and *TNFAIP6*, which had previously been associated with sepsis in other research. Using these genes, we trained a logistic regression classifier to distinguish whether a sample contains an *E. coli* or *S. aureus* infection or belongs to a healthy control group, and subsequently assessed its performance. The classifier achieved an Area Under the Curve (AUC) of 0.96 for *E. coli* and 0.98 for *S. aureus*-induced sepsis, and perfect discrimination (AUC of 1) for healthy controls from the other conditions in a 10-fold cross-validation. The genes demonstrated an AUC of 0.75 in distinguishing between sepsis patients with *E. coli* and *S. aureus* pathogens. These findings were further confirmed in two distinct independent validation datasets which gave high prediction AUC ranging from 0.72–0.87 and 0.62 in distinguishing three groups of participants and two groups of patients respectively. These genes were significantly enriched in the immune system, cytokine signaling in immune system, innate immune system, and interferon signaling. Transcriptional patterns in blood can differentiate patients with *E. coli*-induced sepsis from those with *S. aureus*-induced sepsis. These diagnostic markers, upon validation in larger trials, may serve as a foundation for a reliable differential diagnostics assay.

## Introduction

Sepsis is a life-threatening condition that affects millions of people worldwide and remains a major challenge for healthcare systems [1]. It is defined by the dysregulation of the host's

**Funding:** The author(s) received no specific funding for this work.

**Competing interests:** The authors have declared that no competing interests exist.

inflammatory response due to a microbial infection. The uncontrolled inflammation has the risk to cause damage to tissues and organs, ultimately ending in the death of the patient within hours [2]. Common pathogens associated with sepsis include *Escherichia coli* and *Staphylococcus aureus*, both representing gram-negative and gram-positive bacteria, respectively [1, 3]. Gram-negative and gram-positive bacteria require distinct antibiotic treatments. Furthermore, antibiotics should be administered promptly, ideally within 3–5 hours after the first suspicion of possible sepsis, to decrease the mortality rate [4]. This connection highlights the crucial need for rapid and reliable classification techniques for the timely commencement of an appropriate antibiotic treatment.

Current diagnostic standards for identifying causative pathogens include blood cultures and PCR-based assays. These methods directly detect the presence of bacteria in the bloodstream [5, 6]. Blood cultures primarily concentrate on isolating, identifying, and performing susceptibility tests on the pathogen [5], whereas PCR-based assays specifically target and detect unique nucleic acid sequences of the pathogen [6]. It has been shown that PCR-based assays exhibit a faster turnaround time of several hours compared to blood cultures, which necessitate several days to provide final results [5–7]. Despite a sepsis diagnosis, a causative agent for the infection is often not found using either culture or PCR-based methods [6, 8] as the pathogen may have already been eradicated from the bloodstream while still triggering a dysregulated host immune response. Therefore, an alternative approach involves analyzing the immunological role prompted by the pathogen in the host and deducing the pathogen type based on the transcriptional response to infection. This alternative is valuable as changes in the expression of specific host genes can occur earlier than the detection of the pathogen itself, allowing for quicker diagnosis and intervention.

Various studies have demonstrated differences in the host response to bacterial infections emphasizing the distinction between gram-negative and gram-positive bacteria [9, 10]. Efforts have been made to find reliable candidate genes in host blood for the diagnosis of sepsis caused by these pathogens [11–13]. A commonly employed method for investigating candidate genes associated with sepsis involves profiling gene expression using microarrays [14–17]. Despite the potential benefits, microarrays have not been widely adopted in clinical settings for sepsis diagnosis due to cost constraints and a relatively complicated workflow, although microarrays are utilized for other clinical applications such as SNP and copy-number variation analyses [18–21]. Nonetheless, the cost of diagnostics is rarely a determining factor for patients in critical care or emergency wards, where immediate medical attention takes precedence over financial concerns. However, this approach is particularly advantageous as the expression intensities of candidate genes can be subsequently measured using the PCR method for clinical application when the blood culture proves unsuccessful [22].

Gene expression profiles are typically characterized by high dimensionality, with tens of thousands of genes measured simultaneously. Approaches such as integrated analysis for identifying prevalent genes [23], employing Bayesian sparse factor models [14], and refined differential gene expression analysis [24] often yield a large number of gene lists in the output of expression analysis tools. Therefore, it is impractical to incorporate all of these genes for clinical applications. Despite the demonstrated differences in host responses to bacterial infections, the lack of a practical and efficient set of markers hinders the seamless identification and differentiation of gram-negative and gram-positive bacterial pathogens in clinical settings. Addressing this gap in knowledge is crucial for enhancing diagnostic precision and tailoring effective treatments for patients with sepsis caused by *E. coli* and *S. aureus* and reducing antibiotic resistance.

The age risk factor plays a significant role in the severity of the illness and the likelihood of recovery [25, 26]. Studies have shown differences between adults and pediatric septic patients

[27–29]. This emphasizes the importance of personalized approaches in the diagnosis and management of sepsis, as the presentation and underlying causes of the condition can vary significantly between these distinct patient populations [30].

In the current study, we constructed a model based on; Least Absolute Shrinkage and Selection Operator (Lasso) regression technique [31], to identify the set of genes that can differentiate between *E. coli*-, *S. aureus*-induced sepsis, and healthy cases in adult patients.

Lasso is a regularization technique that adds a penalty term to the loss function of linear regression, which encourages the model to select only the most important genes and shrink the coefficients of the less important genes to zero. By using Lasso to select genes, we can reduce the dimensionality of the gene space and improve the performance of the classification model [31].

Following selecting the most important genes using Lasso, we trained the logistic regression model on the selected genes to predict the class of the testing set. The logistic function maps any real-valued input to a probability between 0 and 1, which can be interpreted as the predicted probability of the positive class [32]. A classifier utilizing these markers permits the categorization of responsible microorganisms in new samples. These findings were further validated in external datasets containing patients with community-acquired sepsis infections. We assessed the model's internal validation and stability and then investigated Gene Ontology (GO) enrichment and Reactome pathway analysis. In this context, our study identified predictive genes differentiating between *E. coli* or *S. aureus* sepsis in adults, offering a promising approach for the diagnosis and management of sepsis caused by these pathogens.

## Materials and methods

### Microarray dataset

We utilized the publicly available GEO database (www.ncbi.nlm.nih.gov/geo/), which contains high-throughput gene expression data, to identify relevant studies related to sepsis in *Homo sapiens*. Our search in GEO was conducted using specific keywords such as "sepsis", "*S. aureus*", "*E. coli*", and "array". Studies that have used the microarray technique, whole blood samples; involved adult sepsis patients with positive culture results for both *E. coli* and *S. aureus* as well as normal controls were included in this study. One dataset, with an accession number of GSE33341 [14], met the selection criteria (15 April 2023) and was consequently retained for further analysis. The dataset includes patients with confirmed bloodstream infections (BSI), comprising 19 cases of *E. coli*-induced, 32 cases of *S. aureus*-induced sepsis, and 43 healthy, uninfected control samples (Table 1). Individual participant's identification was not feasible during or after data collection.

### Data preparation

The primary data for the study with accession number GSE33341 [14] was obtained from the GEO database and analyzed using R software (v4.4.0) [33] and Bioconductor packages [34]. Normalization of Affymetrix microarrays was conducted using Robust Multichip Average (RMA) [34]. For input, all transcripts that were detected in at least one sample were used, without screening for differential expression. Following the conversion of the probe ID to unique official gene symbols, the dataset was subjected to a transformation process resulting in the arrangement of the 94 samples in rows and 22,277 genes in columns.

**Table 1. Cohort characteristics by age and gender.**

| Parameters | *E. coli* | *S. aureus* | Control |
|---|---|---|---|
| **Sex, n** | | | |
| Female | 14 | 6 | 21 |
| Male | 5 | 26 | 22 |
| **Age range (median), year** | | | |
| Female | 25–91 (43) | 36–75 (53) | 21–53 (26) |
| Male | 40–70 (43) | 24–91 (40.5) | 23–59 (25) |
| **Age group, n** | | | |
| Female | | | |
| Age < 65 | 8 | 3 | 21 |
| Age > 65 | 6 | 3 | 0 |
| Male | | | |
| Age < 65 | 4 | 19 | 22 |
| Age > 65 | 1 | 7 | 0 |
| **Age-gender upsampled, n\*** | | | |
| Female | 16 | 38 | 21 |
| Male | 16 | 38 | 22 |
| **Group upsampled, n\*** | 76 | 76 | 76 |

\*Refer to the section 'Upsampling for group balance'

## Model selection and imbalance mitigation strategies

To ascertain the appropriate model for identifying gene markers which classify *E. coli-* and *S. aureus*-induced sepsis and healthy controls, we employed three well-established machine learning algorithms: Ridge [35], Elastic Net [36] and Lasso regression [31]. Briefly, utilizing normalized expression data, we conducted model construction by partitioning the dataset into training (80%) and testing (20%) sets. Through tenfold cross-validation with three repeats, we fine-tuned the lambda hyperparameters across the models. Subsequently, Lasso regression emerged as the most suitable model for the transcriptome data and was utilized for further testing, as it maintained comparable performance while selecting fewer genes compared to other methods (S1 File). Detailed steps for the application of Lasso regression are delineated in the following section, with identical configurations applied for Ridge and Elastic Net regression. Furthermore, in order to address the effects of imbalanced data during model training, we explored different upsampling techniques, including the Synthetic Minority Oversampling Technique (SMOTE) [37] and random upsampling specifically for minority groups, implemented on the normalized expression data before parameter tuning. We then reassessed the Ridge, Elastic Net and Lasso models with the same configuration as previously on the upsampled data. Following evaluation of mean square error (MSE) metric, random upsampling of minority groups was identified as the most suitable method for our dataset and was subsequently utilized throughout our study (S1 File). Further elaboration on the methodology employed for random upsampling will be presented in subsequent sections of this paper.

## Predictive genes discovery

In our study, we utilized two approaches of unsupervised learning using Principal Component Analysis (PCA); to reduce the dimensionality of our dataset and visualize the relationship between samples in a reduced dimensional space. Moreover, we used a supervised learning

method; Lasso regression [31], to select genes that could differentiate between *E. coli- and S. aureus*-induced sepsis. To ensure the robustness of the analysis, 10-fold cross-validation with three repeats was performed while maintaining the same proportion of each group at each fold. A penalty was applied, and the penalty constant was determined by tuning the lambda hyperparameter within the range of 0 to 1, with a step size of 0.01. The Lasso model was employed with a fine-tuned lambda parameter for the selection of genes. To predict the performance of the Lasso model, the dataset was split into two sets of training and test sets, with each set consisting of 80% and 20% of the samples, respectively. A Lasso regression model was generated using the training dataset with a tuned lambda value. Subsequently, the model's predictive capability was assessed using the MSE metric. Furthermore, we investigated the variations in MSE induced by Lasso in both smaller and larger sample groups. To study the smaller sample size, we utilized the training dataset, applied robustness through cross-validation with the same configurations, and determined the lambda parameter with tuning as described. Subsequently, the dataset was partitioned into training and test sets with 80% and 20% of samples respectively, and the Lasso model was built using the training dataset, followed by an examination of the MSE. For the larger sample data, we employed an upsampling technique, as described in the following section. We maintained the same configuration for cross-validation, penalty constant tuning, Lasso model building, and prediction as detailed previously.

Through Lasso regression, a set of predictive genes were selected. These genes were further evaluated for predictive performance using logistic regression [32] and the area under the curve (AUC) and precision, recall, F1-score, and accuracy metrics. Moreover, we applied hierarchal clustering to discover distinct gene clusters and their underlying pattern within our predictive genes.

## Upsampling for group balance

Age over 65 and gender have been reported to be a risk factor for developing sepsis [25, 26, 38, 39]. To further explore the stability of the predictive genes, we stratified the samples by age and gender, creating four subgroups: young male ($< 65$), young female ($< 65$), old male ($> 65$), and old female ($> 65$). An imbalance in sample sizes across these subgroups was observed, with the old male subgroup having the fewest samples (Table 1). To mitigate the effect of this imbalance, we used an upsampling technique.

In the first step, we addressed age imbalance by randomly duplicating existing samples in the minority age subgroups using 'resample' method until the number of samples in each age subgroup was equal. Secondly, the gender imbalance was addressed by randomly upsampling existing samples in the minority gender subgroup. As a result, we obtained an equal number of female and male samples within each bacterial group with a total of 151 samples and 22,277 genes (Table 1), which allowed us to train and test our predictive genes using the same number of age-gender balanced samples for each group. We did not perform age-gender-based upsampling for the healthy control group as the subgroups were already balanced.

In the third step, the imbalance between the bacterial and the healthy control groups was addressed by randomly upsampling the minority groups of *E. coli* and healthy control to match the number of samples in the *S. aureus*-induced sepsis group. This resulted in a total of 228 samples with 22,277 genes (Table 1), allowing us to train and test the predictive genes using an equal number of samples for each group.

## External validation in independent datasets

To ensure the generalizability of our results, we validated our predictive gene set using two additional datasets, GSE13015 [40] and GSE65088 [17]. In GSE13015, we examined data from

4 *E. coli* (female, ages 63–72), and 2 *S. aureus* (female, ages 52–74) samples of sepsis patients, and 5 healthy control samples (2 females and 3 males, ages 35–67), all obtained from whole blood and RNA expression levels were assessed using the Illumina HumanHT-12 v3 Expression BeadChip array (Illumina, San Diego, CA, USA). Similarly, GSE65088; a BSI resembling dataset, consisted of 10 *E. coli* and 10 *S. aureus* samples cultured in whole blood samples from healthy donors (male, age $\geq$ 40 years) and analyzed using the Illumina HumanHT-12 v4 Expression BeadChip array (Illumina, San Diego, CA, USA). For consistency, we normalized both datasets using the quantile method of normaliseIllumina within the beadarray package (v2.52.0) [41] in R (v4.4.0) [33]. In GSE13015, pre-normalization involved background correction as described in Schmid *et al.* [42] using the bgAdjust.affy method of lumiB() function, Lumi package (v2.54.0) [43], enabling subsequent log2-transformation of expression values. Following the conversion of probe IDs to unique official gene symbols, the datasets underwent a transformation process. The intersection of each dataset with the set of predictive genes was then determined. We averaged the expressions of genes with multiple annotations in the validation datasets.

For validation purposes, we utilized GSE13015 to assess the predictive performance of the selected genes in distinguishing between the three groups: *E. coli*-, *S. aureus*-induced sepsis, and healthy control. In parallel, GSE65088 was leveraged to validate the predictive performance of the selected gene model in discriminating between *E. coli*- and *S. aureus*-induced samples. The efficacy of our gene-based diagnostic approach was evaluated using the AUROC curve as the primary metric.

## Protein-Protein Interaction (PPI) network creation

After identifying the predictive genes using the Lasso regression algorithm, a protein-protein interaction (PPI) network was constructed by integrating the data with the STRING database [44]. The Cytoscape software (v3.9.1) [45] was used to visualize the interactive relationship between the predictive genes. To identify densely connected regions in the network, the Molecular Complex Detection (MCODE) plug-in of Cytoscape was utilized to cluster the proteins [46].

## Functional annotation analysis of candidate genes

To investigate the biological significance of the predictive genes, functional and pathway enrichment analyses were conducted using the PANTHER (Protein ANalysis THrough Evolutionary Relationships) [47], and Reactome [48] resources within the Gene Ontology (GO) interface [49–51]. An FDR *p*-value <0.05 was considered statistically significant for enrichment.

## Statistical analysis

For all statistical tests, we utilized Jupyter Notebook v6.0.3 [52], the Scikit-Learn [53], and the Seaborn library [54]. We employed the Mann-Whitney test (two-sample Wilcoxon test) to assess the differences in gene expression levels between the *E. coli*- and *S. aureus*-induced sepsis groups as needed. The Pearson correlation coefficient was utilized to measure the linear association between predictive genes.

## Results

### Identification of predictive genes

To identify the predictive genes within the pool of 22,277 genes associated with the diagnosis of *E. coli-* and *S. aureus*-induced sepsis in 94 samples, we employed Lasso regression with a lambda value of 0.07. The Lasso regression model yielded a nonzero coefficient for 25 genes (S1 Table) which exhibited an MSE of 0.20, implying an 80% predictive accuracy.

To investigate the variation of MSE, we employed Lasso regression to a more restricted dataset, specifically the training dataset with a sample size of n = 76, using the same cross-validation parameters, a lambda value of 0.07 was achieved, resulting in an MSE of 0.15. The Lasso model identified 20 genes, with 14 genes overlapping the 25 selected genes (S1 Table).

Expanding the investigation to include a larger sample size through upsampling to 151-sample and 228-sample datasets, Lasso selected all available genes with a lambda parameter set to 0 in both conditions. The MSE for the 151-sample dataset reduced to 0.07, while for the 228-sample dataset, it further decreased to 0.04. This underscores the heightened predictive performance of the gene selection model when provoked with a more extensive dataset obtained through upsampling.

### Classification analysis of three groups of participants

To assess the predictive performance of the selected genes in classifying the three groups of *E. coli-*, *S. aureus*-induced sepsis, and healthy controls, we established a logistic regression model utilizing the 25 selected genes. Subsequently, we generated receiver operating characteristics (ROC) curves to measure the effectiveness of these predictive genes, as illustrated in Fig 1A. The 25-gene model exhibited an AUC of 0.96 when distinguishing *E. coli*-induced sepsis from other conditions, and an AUC of 0.98 for *S. aureus*-induced sepsis from other circumstances. The model achieved a perfect AUC of 1 for distinguishing healthy controls from other groups, indicating perfect separation of the healthy group from sepsis patients (metrics available in S2 Table).

The 25 predictive genes were analyzed by group (*E. coli* vs *S. aureus* vs healthy control) using a PCA plot (Fig 1B). The PCA plot indicated the clustering of samples based upon groups, with the separation of healthy controls from *E. coli-* and *S. aureus*-induced sepsis cases. Partial overlap between *E. coli-* and *S. aureus*-induced sepsis, particularly along PC2 and PC3 indicates some similarities in gene expression between these two patient groups or that there is some variability within each group. Furthermore, PC3 could indicate a subpopulation within the three groups of *E. coli-*, *S. aureus*-induced sepsis, and healthy controls.

### Internal validation of the predictive genes

To validate the outcomes of Lasso feature selection, we assessed the distribution and performance of the selected 25 genes, aiming to determine the distributional properties of genes between samples. Skewness and kurtosis metrics were employed as indicators of symmetry and peakedness, respectively, with the normal range defined as -2 to +2 for skewness and -7 to +7 for kurtosis [55, 56]. The predictive genes showed positive and negative skewness within the range of +/-2 for *E. coli*-induced sepsis patients, except for the genes *EIF1AY*, *APOBEC3B*, with positive and *GUSBP3* (also known as *SMA3*) with negative skewness, and positive or limited negative kurtosis within the range of +/-7, except for *GUSBP3* with considerable positive kurtosis. The predictive genes for *S. aureus*-induced sepsis patients demonstrated positive and negative skewness and kurtosis within the range of +/-2 and +/-7, respectively (Fig 2A and 2B).

A

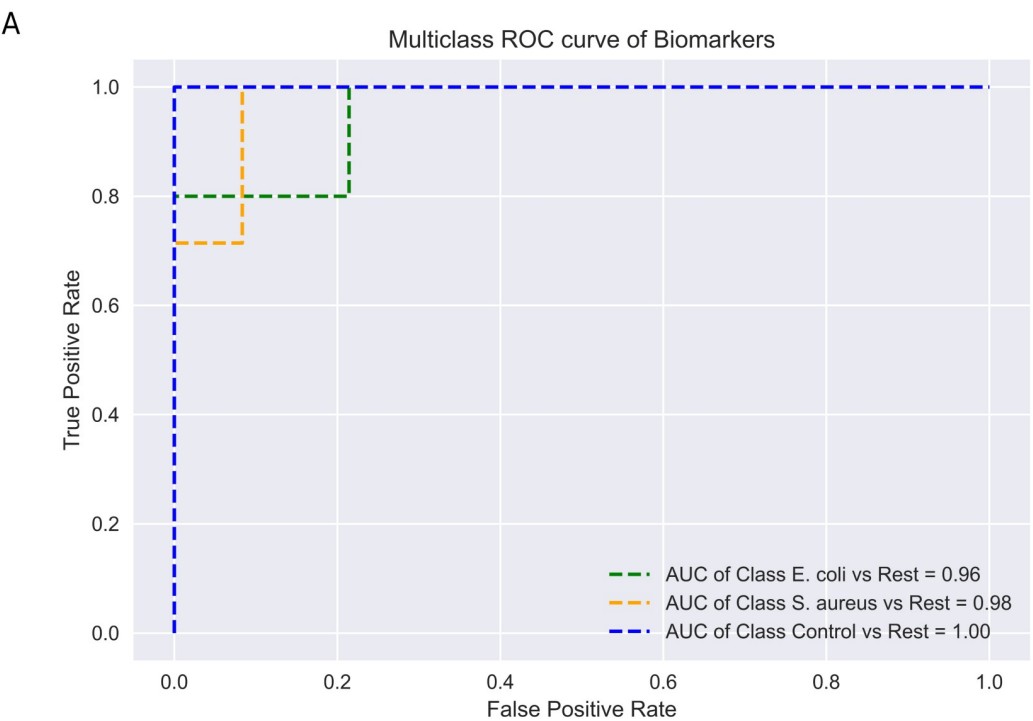

B

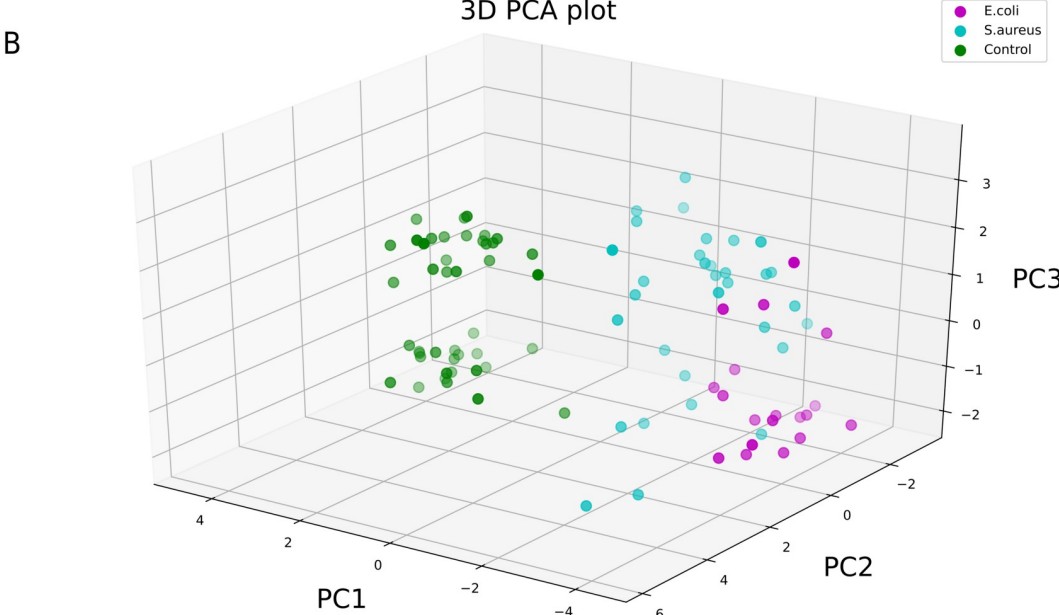

**Fig 1. Assessment of the performance of the selected genes.** (A) Receiver operating characteristic (ROC) curves (*E. coli*; green dot line, *S. aureus*; orange dot line, healthy control; blue dot line) (B) Principal Component Analysis (PCA) plot for the 25-gene model predicting *E. coli*- and *S. aureus*-induced sepsis in the patient (*E. coli*; purple, *S. aureus*; light blue, healthy control; green). The selected genes showed a high predictive performance for classifying each group from other conditions.

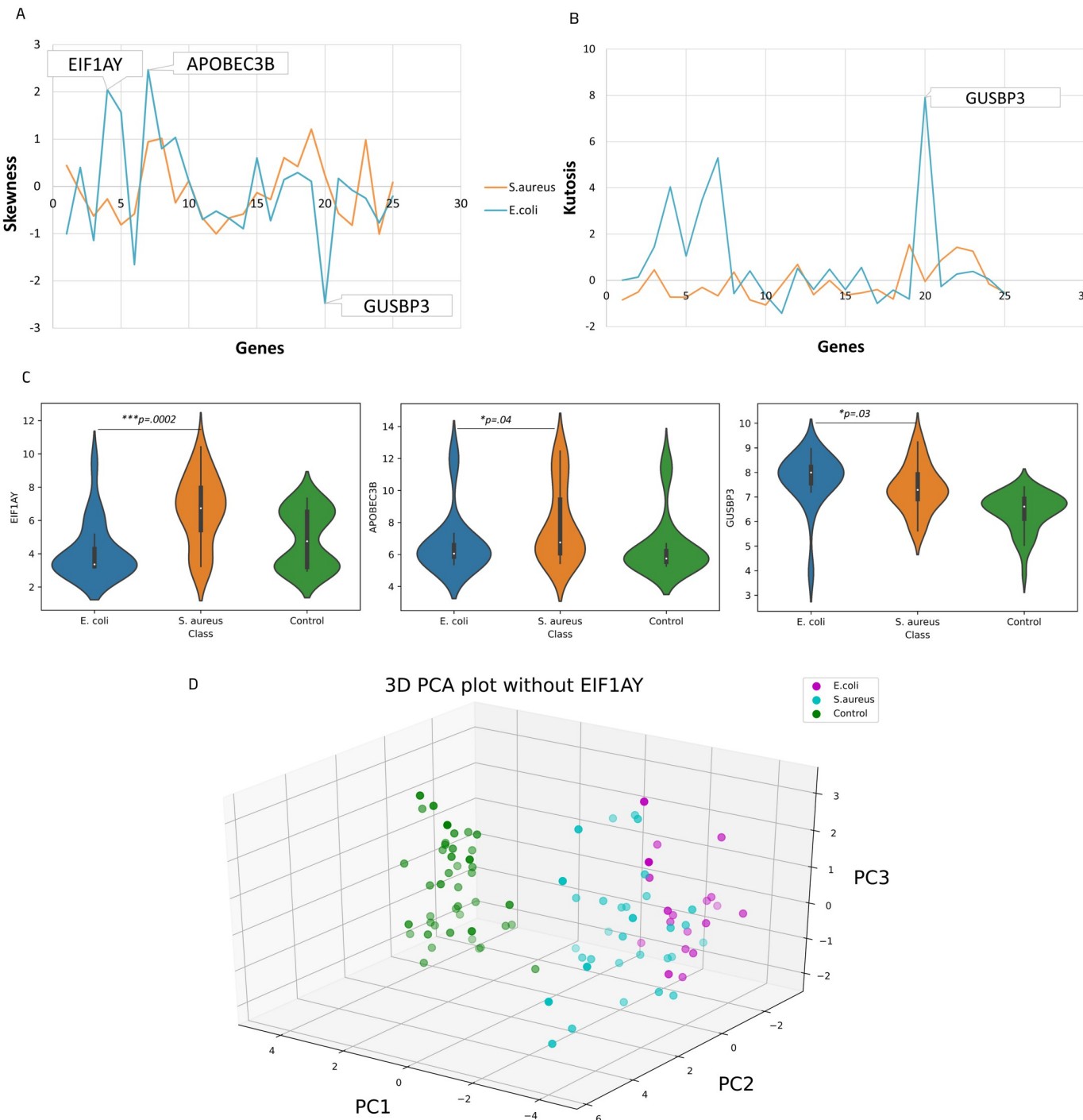

**Fig 2. Internal validation of selected predictive genes.** (A) Skewness and (B) Kurtosis of 25 genes selected by Lasso regression (*E. coli*; blue line, *S. aureus*; red line). (C) Violin plot of three skewed predictive genes in patients with *E. coli*- and *S. aureus*-induced sepsis, as assessed by Mann-Whitney *U* test. (D) PCA plot of predictive genes after removing the *EIF1AY* gene (n = 24). *EIF1AY*: Eukaryotic Translation Initiation Factor 1A Y-Linked, *APOBEC3B*: Apolipoprotein B MRNA Editing Enzyme Catalytic Subunit 3B, *GUSBP3*: Glucuronidase, Beta Pseudogene 3. The predictive genes were robust within the range of +/-2 and +/-7 for skewness and kurtosis respectively.

We performed a Mann-Whitney *U* test to evaluate the differential expression of the three genes (*EIF1AY*, *APOBEC3B*, and *GUSBP3*) in *E. coli-* and *S. aureus*-induced sepsis groups. The corresponding *p*-values for *EIF1AY*, *APOBEC3B*, and *GUSBP3* were 0.0002, 0.038, and 0.027, respectively, with average log2-fold changes of 1.57, 1.16, and 0.95. These results suggest that the observed differences in gene expression between the patients with *E. coli-* and *S. aureus*-induced sepsis are statistically significant (Fig 2C).

Following filtration of the three genes *EIF1AY*, *APOBEC3B*, and *GUSBP3* from the list of 25 predictive genes and reassessment of the model in classifying two patient groups and healthy control, the AUC increased to 1 for all groups, with the same accuracy, precision, recall, and f1-score as before the filtration of these genes from the dataset (S2 Table). This indicates that the three genes in *E. coli*-induced sepsis may have had a negative impact on the performance of the model. This finding was also supported by the PCA analysis which demonstrated an improvement in PC1 (38.40%), and a decrease in PC3 (6.69%) after filtration of these genes (S2 Table). As a result of the filtration of the three genes, the subpopulation within each group was less pronounced, with the PCA plot demonstrating that *EIF1AY* played a key role in this modification (Fig 2D). As it can be inferred from Fig 2C and Table 1, the gene was expressed in the female group with *E. coli*-induced sepsis.

## Classification analysis of two groups of patients

To evaluate the performance of the 25 and 22 predictive genes (unfiltered vs filtered genes of *EIF1AY*, *APOBEC3B*, and *GUSBP3*) in classifying samples from the two patient groups, *E. coli*-induced sepsis vs. *S. aureus*-induced sepsis, we conducted a two-group classification analysis using logistic regression algorithm. The AUC values for the 25-gene set and the 22-gene set were 0.75 and 0.89, respectively. The higher AUC value obtained with the 22-gene set suggested improved performance in discriminating between the two patient groups (Fig 3A). Furthermore, we performed a hierarchical heatmap clustering using Ward's method on 25 predictive genes to discover gene expression patterns. The analysis led to the formation of two distinct gene clusters (Fig 3B) suggesting unsynchronized behavior in response to the *E. coli* and *S. aureus* pathogens between clusters. This was further supported by the lack of significant correlations between genes, as evidenced by the prevalence of values in the low positive range (0 to 0.25) and low negative range (-0.25 to 0) in Pearson correlation analysis (S1 Fig). This suggests genes do not exhibit a strong association with each other, aligning with their high predictive performance.

## Stability analysis of predictive genes

To address dataset imbalance, we employed upsampling techniques, ensuring 16 samples per gender subgroup for *E. coli*-induced sepsis and 38 for *S. aureus*-induced sepsis, minimizing age-gender discrepancies (Table 1, S2A Fig). For broader predictive gene consistency, we evenly distributed participants among *E. coli*, *S. aureus*, and healthy control groups, resulting in 76 samples per group (Table 1). Stability assessment of predictive genes using PCA, AUC, and classification metrics demonstrated improved separation of *E. coli-* and *S. aureus*-induced cases (S2A and S2C Fig, S2 Table) with an AUC of 1 for all groups (S2B and S2D Fig).

## Diagnostics power assessment of 25-gene classifier

To further assess the diagnostic efficacy of the model in distinguishing between *E. coli-* and *S. aureus*-induced sepsis, we applied the AUROC curve analysis to both validation datasets. In the first validation dataset (GSE13015), 23 genes overlapped with the 25-gene model (except for *IGHG1* and *YLPM1*, 92% identity). As shown in Fig 4A, using the 23 shared genes, the model demonstrated strong discriminative performance, with an AUC of 0.79 for *E. coli*-

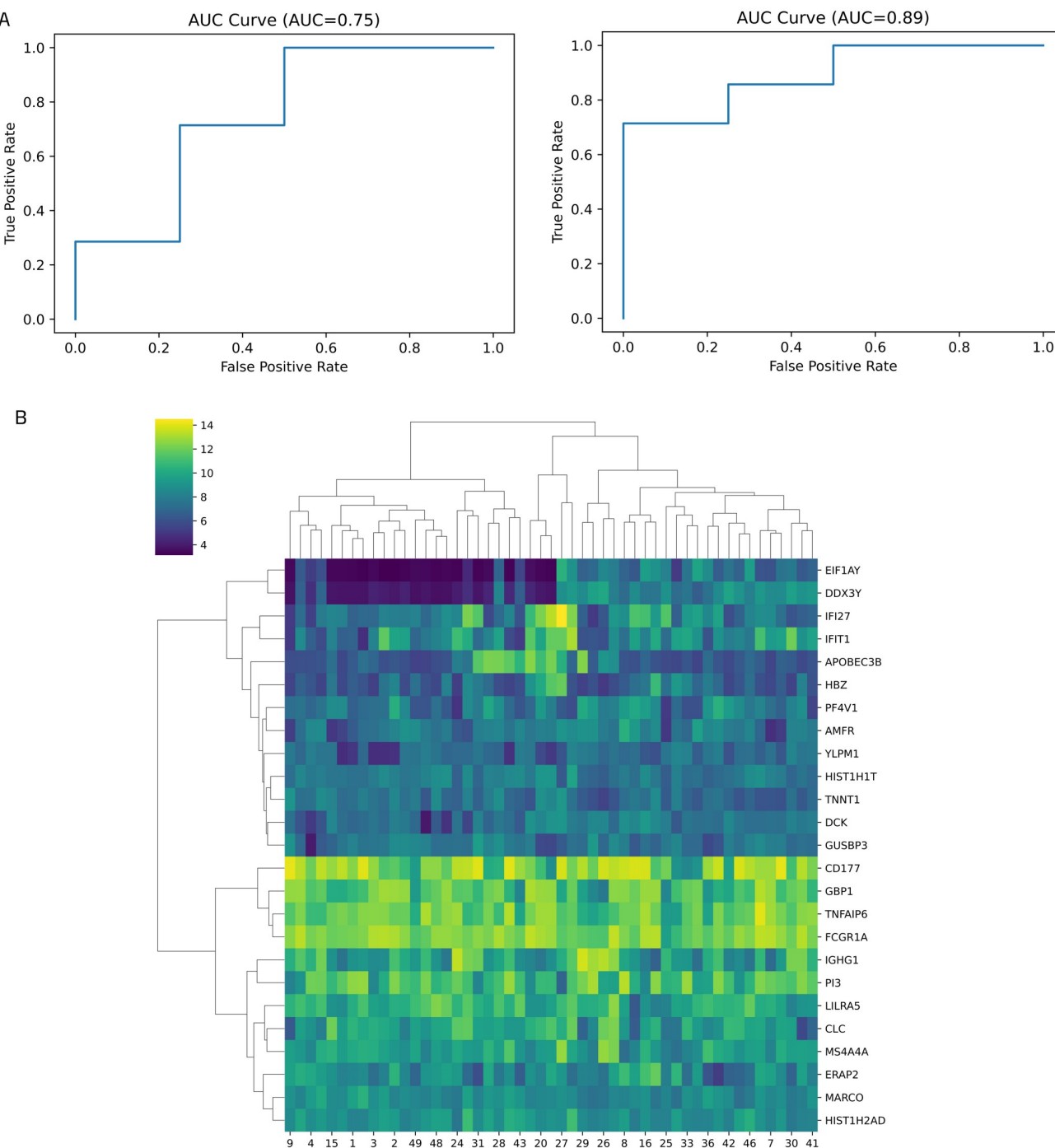

**Fig 3. Classification performance of *E. coli*- and *S. aureus*-induced sepsis.** (A) AUROC curves generated using the 25-gene model (left) and the 22-gene model (right) (B) Hierarchical clustering of *E. coli*- and *S. aureus*-induced sepsis with the 25-gene model. The selected genes showed high predictive performance for classifying two groups of patients.

induced sepsis cases, 0.72 for *S. aureus*-induced sepsis cases, and a notable AUC of 0.87 for healthy control cases. In the second validation dataset (GSE65088), we found 24 overlapped genes (except for *IGHGI*, 96% identity) which distinguished *E. coli* and *S. aureus* infection cases with an AUC of 0.62 (Fig 4B).

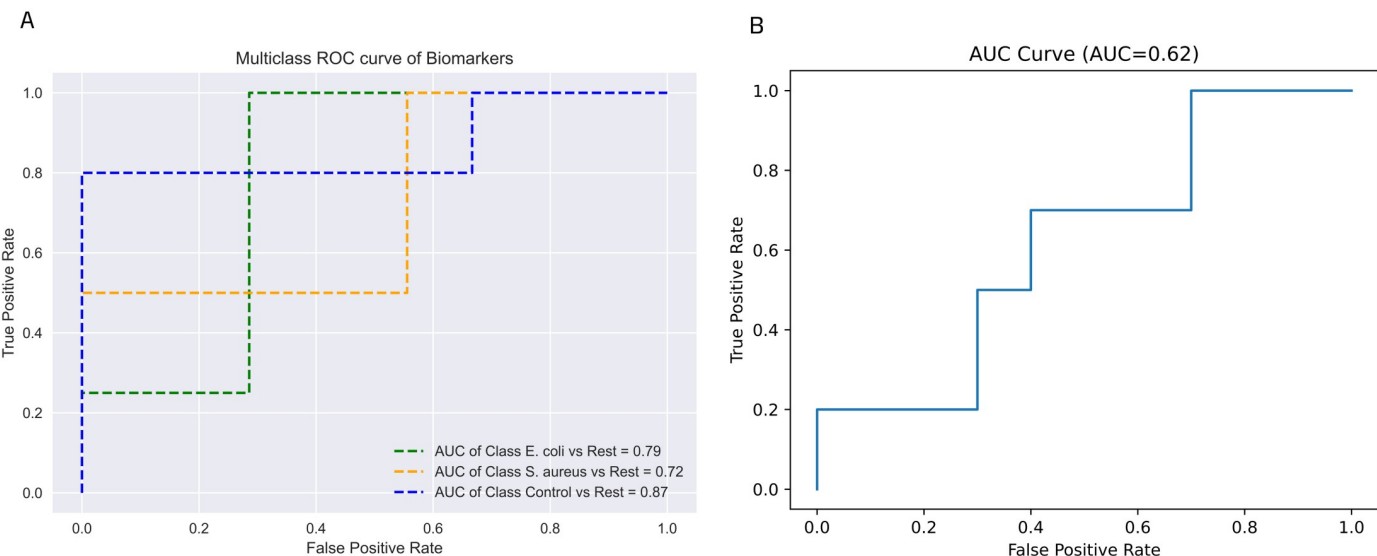

**Fig 4. Model performance in two distinct scenarios using AUROC.** (A) Distinguishing between *E. coli*-, *S. aureus*-induced sepsis, and healthy control using the GSE13015 dataset, and (B) Separating *E. coli* and *S. aureus* infections using the GSE65088 dataset. The selected genes showed high diagnostic performance in distinct external datasets for classifying the three groups of participants.

## Integrative analysis of gene associations

Analyzing associations among these genes could reveal mechanistic differences between *E. coli*- and *S. aureus*-induced sepsis. Using STRING [44], the PPI network analysis on proteins corresponding to the 25 genes revealed a sparse network (23 nodes, 6 edges) with a protein-protein interaction enrichment *p*-value of < 0.0187 for 8 proteins (IFIT1, IFI27, GBP1, FCGR1A, EIF1AY, DDX3Y, HIST1H1T (H1-6), and HIST1H1AD (H2AC7)), indicating partial functional or physical connections (Fig 5A). In Cytoscape with the MCODE plug-in, we identified a densely connected protein cluster (IFI27, IFIT1, GBP1, and FCGR1A) (Fig 5B) and noted functional associations among EIF1AY, and DDX3Y, as well as HIST1H1T and HIST1H1AD. However, other proteins showed no associations, indicating that genes associated with each bacterium may activate distinct pathways. Concurrently, we conducted a functional enrichment analysis on the 25 predictive genes using the GO interface and PANTHER resource. In the GO ontology analysis, the categories within the biological process were notably enriched in immune system processes, defense responses, and immune responses (S3A Table). Moreover, Reactome pathway analysis revealed immune system-related pathways including cytokine signaling, innate immune system, and interferon signaling as the most significant pathways associated with the predictive genes (S3B Table).

## Discussion

The current study is designed to identify transcriptional markers distinguishing between *E. coli*-, *S. aureus*-induced sepsis, and healthy control in adult patients. We employed a data-driven approach incorporating the Lasso regression algorithm for the selection of genes and logistic regression for prediction. The algorithm was applied to a dataset comprising septic patients with positive blood cultures and healthy control cases (GEO accession number GSE33341).

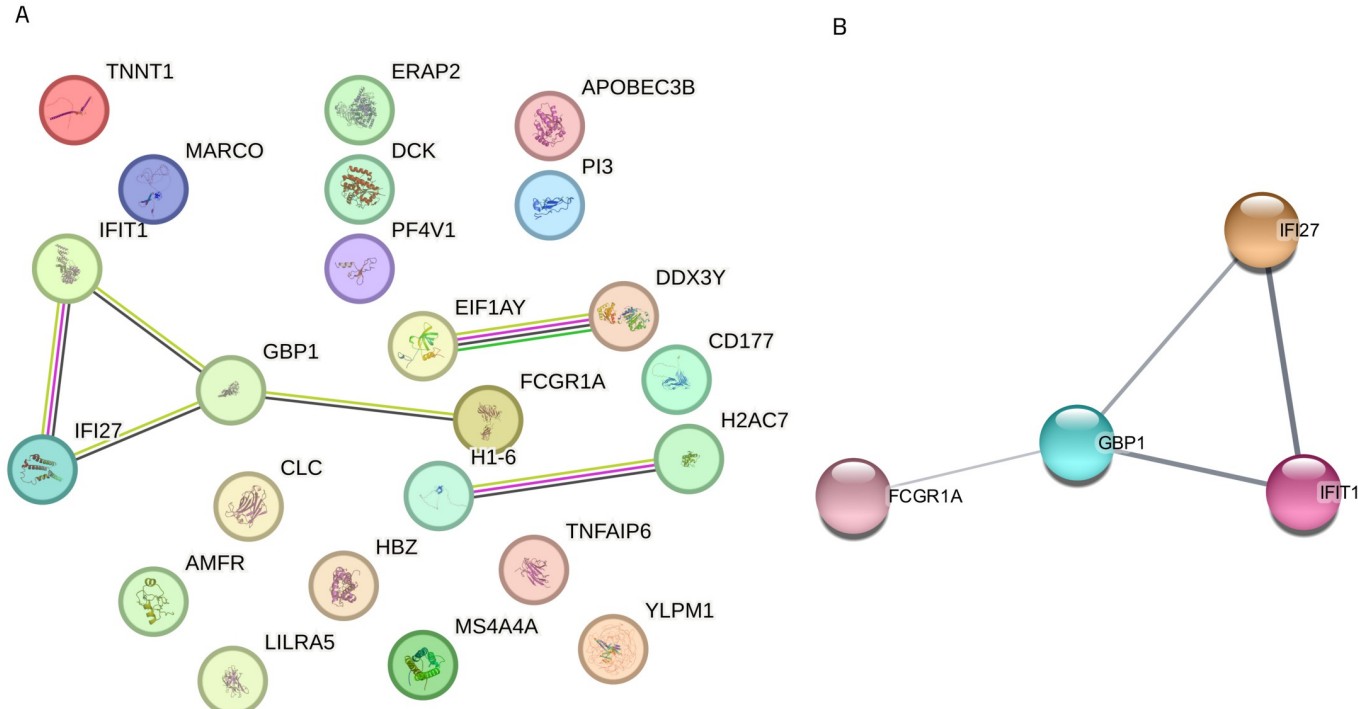

**Fig 5. Networking and clustering of predictive genes.** (A) Protein-Protein interaction network of 25 predictive genes using STRING (B) One cluster detected by Cytoscape consisting of 4 nodes and 4 edges. The lack of significant association between the selected genes supports the high predictive performance in the classification of the groups.

We identified a 25-gene classifier that was strongly associated with the patient groups, indicating their high informativeness in distinguishing sepsis induced by *E. coli* or *S. aureus* from other conditions. Our findings were supported by the high AUC values obtained in the ROC analysis, which demonstrated that the 25-gene model had excellent accuracy in predicting sepsis induced by *E. coli* (AUC of 0.96) or *S. aureus* (AUC of 0.98) pathogens from other groups. Notably, the AUC values were consistent with those reported in the Ahn *et al.* study, where the factor classifier distinguished *E. coli*-induced sepsis from healthy controls with an AUC of 0.92, and *S. aureus*-induced sepsis from healthy controls with an AUC of 0.9898 [14].

In our analysis, we also found that the 25-gene model had an acceptable AUC for predicting *E. coli*- and *S. aureus*-induced sepsis (AUC of 0.75), which further improved to an excellent AUC after filtering out three differentially expressed genes (AUC of 0.89). This AUC value was similar to the AUC reported by the Bayesian sparse factor classifier (AUC: 0.8503) [14].

Imbalance datasets can pose a challenge for machine learning algorithms because they tend to be biased towards the majority class, resulting in poor performance on the minority class [57]. To address this problem, one approach is upsampling, which involves increasing the number of instances in the minority class. This can be done through techniques such as random duplication. The goal of upsampling is to balance the distribution of classes in the dataset, allowing the algorithm to learn from the minority class more effectively [58]. The results of our study demonstrate the robustness of the 25-gene classifier in predicting *E. coli*- and *S. aureus*-induced sepsis, as evidenced by its stability upon upsampling.

We assessed the 25-gene model using an additional dataset consisting of whole blood samples from adults with *E. coli*-, *S. aureus*-induced sepsis, and healthy control cases (GSE13015).

The robust AUC values of 0.79 for *E. coli*, 0.72 for *S. aureus*, and 0.87 for healthy control identification in our first validation dataset strongly affirm the reliability of the 25-gene classifier.

Furthermore, we extended our evaluation to a separate dataset resembling BSI; one stage before sepsis (GSE65088), resulting in an AUC of 0.62 for distinguishing *E. coli* and *S. aureus* infection. Despite the variations in the number of available genes (23 in the first dataset and 24 in the second), differences in sample size (smaller in the validation datasets compared to training), and variations in dataset homogeneity, the model's performance remained commendable.

In a 2019 cross-sectional study conducted by Chen *et al.* [23] transcriptional biomarkers with distinct expression patterns in *E. coli*- and *S. aureus*-induced sepsis were investigated. Notably, *LILRA5* and *TNFAIP6* were among the top 9 genes and were also part of the 25-gene classifier identified in our current study. *LILRA5*, a leukocyte immunoglobulin-like receptor, plays a role in modulating immune responses through its interactions with different ligands. It has been suggested to have a role in the recognition of both viral and bacterial pathogens, and in regulating the inflammatory response [59–61]. *TNFAIP6*, Tumor Necrosis Factor Alpha-Inducible Protein 6, has also been shown to play a role in inflammation and immunity. It can regulate the activity of immune cells, such as macrophages, by modulating the production of pro-inflammatory cytokines and chemokines [62].

Sepsis is primarily triggered by the complex interaction between the innate and adaptive immune systems, which involves a delicate balance of pro- and anti-inflammatory cytokines [63]. The main aim of this study was to develop predictive models, rather than providing an in-depth exploration of sepsis pathophysiology. However, our pathway analysis of 25 genes has revealed that they are significantly involved in immune system response and cytokine signaling pathways. These findings suggest that the genes play a role in the body's 'general alarm' response to infection in a broad sense. This could explain the overlap between the *E. coli*- and *S. aureus*-induced sepsis in PCA analysis. However, analysis using hierarchical clustering, Pearson correlation, and pathway analysis has also revealed that different pathways contribute to the response to each bacterial group, indicating a more specific response to each type of bacteria. In other words, our findings support the idea that the body's response to sepsis is composed of two levels: a general response triggered by the immune system's delicate interplay, and a more specific response that is activated by each bacterial group. This two-level response has been suggested by other study as well [23], and our findings provide additional support for this concept.

An increasing amount of evidence indicates that there are sex-related differences in sepsis epidemiology [64–67]. In our investigation, we identified a subpopulation within each group, which softened when gender factors were balanced between the groups. Our findings suggest that the *EIF1AY* gene is responsible for this subgrouping.

The *EIF1AY* gene is typically located on the Y chromosome, which is present only in males [68]. However, our study suggested that this gene is overexpressed in female subjects who have been infected with *E. coli*. The reason for the overexpression of the gene *EIF1AY* in females with *E. coli* infection requires further investigation.

In summary, our study successfully identified a set of predictive genes that can differentiate between *E. coli* or *S. aureus* community-onset sepsis in adult patients, providing a promising approach to the diagnosis and management of sepsis caused by these pathogens. However, validation in larger cohorts is necessary to assess clinical applicability. Given our study's limitations-a small sample size and a straightforward algorithm- future investigations should explore these genes in larger cohorts using more advanced algorithm for deeper insights. Exploring transcriptional profiles in gram-negative and gram-positive bacterial sepsis through microarray analysis opens new avenues for rapid and precise patient diagnosis.

## Supporting information

**S1 Fig. Pearson correlation heatmap in 25 genes in two groups of *E. coli-* and *S. aureus-* induced sepsis.**
(TIF)

**S2 Fig. Stability analysis upon upsampling of age, gender, and groups.** (A) Principal component analysis (PCA) plot and (B) Receiver operating characteristic (ROC) curve of age-gender balanced data (n = 151). (C) PCA plot and (D) ROC curve in group balanced data (n = 228).
(TIF)

**S1 Table. List of genes selected by Lasso regression.**
(DOCX)

**S2 Table. Classification metric and PCA report of selected genes by Lasso regression.**
(DOCX)

**S3 Table. Functional characteristics of selected genes.** (A) Categories of biological process for 25 predictive genes based on the Gene Ontology enrichment analysis (B) Categories of pathways for 25 predictive genes based on Reactome pathway analysis.
(DOCX)

**S1 File. Models and upsampling techniques performances.**
(DOCX)

**S2 File. Sample distribution pattern before feature selection.**
(DOCX)

## Acknowledgments

The author expresses gratitude to Prof. Erik Kristiansson of Chalmers University of Technology for his invaluable technical guidance and constructive reviews, which improved the latest version of the manuscript. The author is thankful to Dr Astrid von Mentzer (University of Gothenburg) for reviewing and coordinating the latest version of the manuscript. Additionally, the author acknowledges Dr Marjan Alirezaee (formerly at Örebro University) for her precious methodological guidance and Dr Soumyadeep Nandi (Umeå University) for his valuable input in interpreting the results and feedback on the earlier version of the study.

To reflect the contribution of the author, the following is provided according to the CRediT Taxonomy: MIS- Project Conceptualization; Data curation; Formal analysis; Funding; Investigation; Methodology; Project administration; Resources; Software; Project oversight; Creation of data visualization and figures and tables; Writing of manuscript.

## Author Contributions

**Conceptualization:** Mahnaz Irani Shemirani.

**Data curation:** Mahnaz Irani Shemirani.

**Formal analysis:** Mahnaz Irani Shemirani.

**Funding acquisition:** Mahnaz Irani Shemirani.

**Investigation:** Mahnaz Irani Shemirani.

**Methodology:** Mahnaz Irani Shemirani.

**Project administration:** Mahnaz Irani Shemirani.

**Resources:** Mahnaz Irani Shemirani.

**Software:** Mahnaz Irani Shemirani.

**Supervision:** Mahnaz Irani Shemirani.

**Validation:** Mahnaz Irani Shemirani.

**Visualization:** Mahnaz Irani Shemirani.

**Writing – original draft:** Mahnaz Irani Shemirani.

**Writing – review & editing:** Mahnaz Irani Shemirani.

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
