## [Decision Letter · Decision Letter 0]

25 Apr 2024

PONE-D-24-01404Transcriptional markers classifying *Escherichia coli* and *Staphylococcus aureus* induced sepsis in adults: a data-driven approachPLOS ONE

Dear Dr. Irani Shemirani,

Thank you for submitting your manuscript to PLOS ONE. After careful consideration, we feel that it has merit but does not fully meet PLOS ONE’s publication criteria as it currently stands. Therefore, we invite you to submit a revised version of the manuscript that addresses the points raised during the review process.

Your paper has been carefully evaluated by three experts in the field and all of them suggested to revise your manuscript. I strongly agree and in particular I totally share the points raised by the reviewer #1.

We look forward to receiving your revised manuscript.

Kind regards,

Vittorio Sambri, M.D., Ph.D.

Academic Editor

PLOS ONE

Reviewers' comments:

Reviewer's Responses to Questions

**Comments to the Author**

1. Is the manuscript technically sound, and do the data support the conclusions?

Reviewer #1: Partly

Reviewer #2: Yes

Reviewer #3: Yes

2. Has the statistical analysis been performed appropriately and rigorously? 

Reviewer #1: No

Reviewer #2: Yes

Reviewer #3: Yes

3. Have the authors made all data underlying the findings in their manuscript fully available?

Reviewer #1: Yes

Reviewer #2: Yes

Reviewer #3: Yes

4. Is the manuscript presented in an intelligible fashion and written in standard English?

Reviewer #1: Yes

Reviewer #2: Yes

Reviewer #3: Yes

5. Review Comments to the Author

Reviewer #1: The methods used by the authors are not between the most advanced available now. For example the LASSO method used for feature extraction has to compared with other methods such as RIDGE and Elastic net, or with methods for dimensionality reduction.

Finally, the numerosity of the data is not so high and the upsampling procedure used by the authors is not convincing.

I would prefer to see the use of synthetic data for data augmentation

Reviewer #2: This article presents an innovative and highly interesting study. I would like to express my gratitude to the authors for designing and conducting this research. It was truly captivating to read. Below, I provide some minor adjustments according to my opinion:

Lines 98-99: Homo sapiens should be italicized.

Line 111: Which version of R? The reference is missing.

Lines 170-173: Illumina: the distribution is missing.

Line 175: The citation for R is missing.

Line 191: Cytoscape software. Version? Reference?

Line 205: Why is a non-parametric test (Mann-Whitney test) used first and then a parametric test (Pearson correlation)? Would it be better to use Spearman correlation?

Line 325: STRING. Citation?

Line 537: "doi" is repeated twice.

In general, would it be possible to increase the resolution of the images (including supplementary ones)? They appear slightly grainy.

Reviewer #3: Dear Dr. Mahnaz,

I read the manuscript with sincere curiosity. The topic of the study is certainly since the introduction of new methodologies for sepsis prediction, and consequently its treatment, is critical. I do not have any major revision to recommend, just a few details that can be improved.

Line 5-6, 33: Never mind specifying the abbreviation given to the name of the bacteria, it is in common use.

Line 57: Do you have considered the costs analysis for the introduction of the microarray in the diagnostic workflow? May the author should add a paragraph about.

Line 98-99: Homo sapiens in italics.

Line 180: what do you meant with “multiple synonyms”?

All over the manuscript: it should be appropriate if the author enter in brackets: the supplier, supplier’s head office and country following each kit and instruments included in the study.

6. PLOS authors have the option to publish the peer review history of their article (what does this mean?). If published, this will include your full peer review and any attached files.

Reviewer #1: No

Reviewer #2: No

Reviewer #3: No

---

## [Author Response · Author response to Decision Letter 0]

29 May 2024

Dear Editor, 

Thanks very much for constructive comments and recommendations. We have addressed all the raised issues and updated the manuscript accordingly. The point-to-point responses to reviewers’ comments is provided in 'Response to Reviewers' file and below, which can be tracked using 'Revised Manuscript with track Changes' file. A ‘Revised Manuscript with Track Changes’ file and a 'Manuscript' file is provided. All images are reproduced with high quality and checked with PACE and uploaded separately. A lab protocol is not applicable for this study. We checked the manuscript with PLOS ONE's style requirements. We removed the the phrase “data not shown”. We hope that the modifications made in the manuscript are satisfying.

Reviewer #1:

The methods used by the authors are not between the most advanced available now. For example, the LASSO method used for feature extraction has to compared with other methods such as RIDGE and Elastic net, or with methods for dimensionality reduction.

Finally, the numerosity of the data is not so high and the upsampling procedure used by the authors is not convincing. I would prefer to see the use of synthetic data for data augmentation. 

We updated the manuscript to include a comparison with Ridge regression, Elastic Net, and oversampling using the synthetic technique SMOTE. 

1. One subsection is added to Materials and Methods section (Lines 147-167)

2. One supporting file added as S1_File 

We have already utilized the dimensionality reduction method Principal Component Analysis (PCA) throughout the study, as stated in lines 168-169 and mentioned in several subsections including ‘Classification analysis of three groups of participants’ and ‘Fig 1’, ‘Internal validation of selected predictive genes’ and ‘Fig 2’, ‘Stability analysis of predictive genes’, ‘S2 Fig’ and ‘S2 Table’. We also updated manuscript by including a supporting information dedicated to PCA before feature selection as S2_File.

Reviewer #2:

Lines 98-99: Homo sapiens should be italicized.

We have changed ‘Homo sapiens’ to ‘Homo sapiens’. 

(Lines 128-129)

Line 111: Which version of R? The reference is missing.

We have added the version and reference. 

(Line 141)

Lines 170-173: Illumina: the distribution is missing.

We have added the distribution. 

(Lines 222 and 225)

Line 175: The citation for R is missing.

We have added both versions to beadarray package and added the reference for R. 

(Line 227)

Line 191: Cytoscape software. Version? Reference?

We have added both version and reference. 

(Line 243)

Line 205: Why is a non-parametric test (Mann-Whitney test) used first and then a parametric test (Pearson correlation)? Would it be better to use Spearman correlation?

We defined criteria for normal distribution, considering skewness and kurtosis, based on the recommendation of Hair et al (2010) [1] and Byrne (2010) [2]. This criterion justifies the methods that we have used. We have clarified this criterion by adding a sentence to the text and updating the sentence. 

(Lines 301-303)

We utilized the Pearson correlation coefficient as a descriptive statistic to measure the strength and direction of the relationship between variables. The parametric aspect of the Pearson correlation arises when conducting hypothesis tests to determine if the observed correlation is significantly different from zero.

Line 325: STRING. Citation?

We have added the reference. 

(Lines 379 and 243)

Line 537: "doi" is repeated twice.

We have removed the extra ‘doi’.

(Line 605)

In general, would it be possible to increase the resolution of the images (including supplementary ones)? They appear slightly grainy.

We updated all images including supplementary. 

Reviewer #3: 

I read the manuscript with sincere curiosity. The topic of the study is certainly since the introduction of new methodologies for sepsis prediction, and consequently its treatment, is critical. I do not have any major revision to recommend, just a few details that can be improved.

Line 5-6, 33: Never mind specifying the abbreviation given to the name of the bacteria, it is in common use.

We have removed abbreviations.

(Lines 30-31 and 58)

Line 57: Do you have considered the costs analysis for the introduction of the microarray in the diagnostic workflow? May the author should add a paragraph about.

We updated the text. 

(Lines 83-88)

Line 98-99: Homo sapiens in italics.

We have changed ‘Homo sapiens’ to ‘Homo sapiens’.

(Lines 128-129)

Line 180: what do you meant with “multiple synonyms”?

In the context of our study, 'multiple synonymous names' refer to genes that are known by more than one identifier or name within the microarray dataset. For instance, some genes listed among the 25-gene dataset were labeled by several names, which may indicate their official symbol, alternative symbols, aliases, or historical names, separated by the '///' sign (i.e., ‘ABCD///EFG///HIJK’). We updated the name to ‘multiple annotations’. 

(Line 233)

All over the manuscript: it should be appropriate if the author enter in brackets: the supplier, supplier’s head office and country following each kit and instruments included in the study.

We have added the distribution. 

(Lines 222 and 225)

1. Hair J, J F, Black JW, Babin BJ, Anderson ER. Multivariate Data Analysis. Seventh ed: Edinburgh: Pearson Education Limited; 2010.

2. Byrne BM. Structural equation modeling with AMOS: Basic concepts, applications, and programming: New York: Routledge; 2010.

---

## [Editor Report · Decision Letter 1]

7 Jun 2024

Transcriptional markers classifying *Escherichia coli* and *Staphylococcus aureus* induced sepsis in adults: a data-driven approach

PONE-D-24-01404R1

Dear Dr. Irani Shemirani,

We’re pleased to inform you that your manuscript has been judged scientifically suitable for publication and will be formally accepted for publication once it meets all outstanding technical requirements.

Kind regards,

Vittorio Sambri, M.D., Ph.D.

Academic Editor

PLOS ONE

---

## [Editor Report · Acceptance letter]

26 Jun 2024

PONE-D-24-01404R1 

PLOS ONE

Dear Dr. Irani Shemirani, 

I'm pleased to inform you that your manuscript has been deemed suitable for publication in PLOS ONE. Congratulations! Your manuscript is now being handed over to our production team.

Kind regards, 

on behalf of

Professor Vittorio Sambri 

Academic Editor

PLOS ONE